# Projection of irrigation water demand based on the simulation of synthetic crop coefficients and climate change

Michel Le Page[1], Younes Fakir[2,3], Lionel Jarlan[1], Aaron Boone[4], Brahim Berjamy[5], Saïd Khabba[2,3], Mehrez Zribi[1]

[1] CESBIO, Université de Toulouse, CNRS/UPS/IRD/CNES/INRAE, 18 Avenue Edouard Belin, bpi 2801, 31401 Toulouse, CEDEX 9, France
[2] Faculty of Sciences Semlalia, Cadi Ayyad University, Marrakech, Morocco
[3] CRSA (Center for Remote Sensing Application), Mohammed VI Polytechnic University, Benguerir, Morocco
[4] CNRM, Université de Toulouse, Meteo-France, CNRS, Toulouse, France
[5] ABHT, Agence du Bassin Hydraulique du Tensift, Marrakech, Morocco

*Correspondance to*: Michel Le Page (michel.le_page@ird.fr)

**Abstract.** In the context of major changes (climate, demography, economy, etc.), the Southern Mediterranean area faces serious challenges with intrinsically low, irregular, and continuously decreasing water resources. In some regions, the proper growth both in terms of cropping density and surface area of irrigated areas is so significant that it needs to be included in future scenarios. A method for estimating the future evolution of irrigation water requirements is proposed and tested in the Tensift watershed, Morocco. Monthly synthetic crop coefficients ($K_c$) of the different irrigated areas were obtained from a time series of remote sensing observations. An empirical model using the synthetic $K_c$ and rainfall was developed and fitted to the actual data for each of the different irrigated areas within the study area. The model consists of a system of equations that takes into account the monthly trend of $K_c$, the impact of yearly rainfall, and the saturation of $K_c$ due to the presence of tree crops. The impact of precipitation change is included in the Kc estimate and the water budget. The anthropogenic impact is included in the equations for $K_c$. The impact of temperature change is only included in the reference evapotranspiration, with no impact on the $K_c$ cycle. The model appears to be reliable with an average r2 of 0.69 for the observation period (2000-2016). However, different sub-sampling tests of the number of calibration years showed that the performance is degraded when the size of the training dataset is reduced. When sub-sampling the training dataset to one-third of the 16 available years, $r^2$ was reduced to 0.45. This score has been interpreted as the level of reliability that could be expected for two time periods after the full training years (thus near to 2050).

The model has been used to reinterpret a local water management plan and to incorporate two downscaled climate change scenarios (RCP4.5 and RCP8.5). The examination of irrigation water requirements until 2050 revealed that the difference between the two climate scenarios was very small (< 2%), while the two agricultural scenarios were strongly contrasted both spatially and in terms of their impact on water resources. The approach is generic and can be refined by incorporating irrigation efficiencies.

## 1 Introduction

Water resources are scarce in semi-arid areas and a major part is allocated to agriculture. In the south-Mediterranean region, irrigation allocation to agriculture represents 80% of total water abstraction. It varies from 46% in eastern countries up to 88% in Morocco in 2010 (FAO, 2016). This percentage has been decreasing in most south-Mediterranean countries during the last decades in particular due to the limitation of available resources and the increase of the urban water demand. In parallel, the pressure on water resources led to what Margat and Vallée (2000) called a "post-dam era" or what Molle et al. (2019) called a "groundwater rush" where subterranean water is overused to satisfy the growing water demand. In recent years, overexploitation of groundwater has been facilitated by technological inventions, affordable cost of exploitation and weak monitoring by authorities (MED-EUWI working group on groundwater, 2007). The overexploitation of aquifers can be observed in different countries over the Mediterranean watershed (Custodio et al., 2016; Le Goulven et al., 2009).

To satisfy the continuous increase in food demand associated with population growth, the agricultural sector has been asked to pursue its already initiated process of conversion towards agricultural intensification and above all towards a sharp increase in yields. This context goes hand in hand with the increaseee in food trade. The replacement of traditional crops by more financially attractive crops is already underway (Jarlan et al., 2016). In the "growth" scenario (which is more or less the actual trend),  presented by (Malek et al., 2018), the annual production of cultivated land increases by 40% and, the production of permanent crops increases by 260%. In the "sustainable" scenario, annual crop production and tree production increase by 30% and 38%, respectively. As a consequence, irrigation water needs are expected to increase (Fader et al., 2016). The expansion and intensification of tree crops will also further rigidify the demand for agricultural water and increase the pressure on groundwater reservoirs (Jarlan et al., 2016) in order to keep tree crops alive during drought events (Le Page and Zribi, 2019; Tramblay et al., 2020). This study is carried on in the Tensift basin in Morocco, where the increase in the irrigated area and the intensification of irrigation during recent decades have caused a long-lasting drop in the groundwater table (Boukhari et al., 2015). A multi-model analysis of the area (Fakir et al., 2015) has shown that the groundwater table falls from 1 to 3 m/year and that the mean annual groundwater deficit (about 100 hm$^3$ since 2000) is equivalent to 50% of the reserves lost during the previous 40 years. Among the main causes of this depletion, is a reduction and higher irregularity of precipitation (Marchane et al., 2017) for crop growth and groundwater recharge, a reduction of snow water storage (Marchane et al., 2015), an increase and intensification of irrigated areas, and a progressive conversion to arboriculture due to national strategy. Since irrigation relies increasingly upon groundwater abstraction, questions are inevitably raised concerning the future of local agriculture and groundwater.

Therefore, in regions where the cropping density and surface area of irrigated areas is growing strongly (see for example the cases of China and India, (Chen et al., 2019)), it is necessary to make projections of the irrigation water demand with the actual trend in order to build alternative scenarios. In their review, March et al. (2012) defined scenario analysis as *internally consistent stories about ways that a specific system might evolve in the future. Scenarios are plausible accounts of the future rather than forecasts*. Narrative scenarios also help identify the drivers of change and the implications of current trajectories

as well as the options for action (Raskin et al., 1998). Scenarios are therefore halfway between facts and speculations in terms of complexity and uncertainty (van Dijk, 2012). They are commonly used as a management tool for strategic planning and for helping managers strengthen decision making.

Various studies (Arshad et al., 2019; Lee and Huang, 2014; Maeda et al., 2011; Schmidt and Zinkernagel, 2017; Tanasijevic et al., 2014; Wang et al., 2016) have addressed the estimation of irrigation water requirements (IWR) of a region with an approach close to equation 1. A simplified balance between crop evapotranspiration ($ET_c$) and effective precipitation ($P_e$) is computed for each crop i and multiplied by the corresponding irrigated area. $ET_c$ can be inferred from the crop coefficient method ($K_c$, Allen et al., 1998). Most of the effects of the various weather conditions are incorporated into $ET_0$ which accounts for the water demand of a reference crop, while the $K_c$ mainly accounts for crop characteristics. $K_c$ varies according to four main characteristics: the crop height, the albedo of the crop and soil, the canopy resistance of the crop to vapor transfer (leaf area, leaf age and condition, and the degree of stomatal control) and the evaporation from soil, especially the soil exposed to solar radiations. Furthermore, as the crop develops, those different characteristics change during the various crop phenological stages. $K_c$ values and stage lengths, for typical climate conditions, are considered to be well known for many crops and have been compiled in the FAO look-up tables.

$$IWR = \sum_{i=1}^{n} (ET_{c_i} - P_e) * Area_i = \sum_{i=1}^{n} \left[ (K_{c_i} * ET_0) - P_e \right] * Area_i \qquad (1)$$

This approach is also a very convenient way to account for both climate and crop changes. On the one hand, $P_e$ and $ET_0$ are obtained from meteorology or climatology. On the other hand, $K_c$ values and stages lengths are taken from the tables. Most of the work consists of evaluating the future irrigated area of each crop and assessing the impact of more efficient irrigation techniques.

There is a significant amount of literature about the estimation of land use and land cover changes (Mallampalli et al., 2016; Noszczyk, 2018), with various techniques to estimate or predict them. Many land cover change approaches are based on transition probability which was introduced by (Bell, 1974) and have been eventually connected to Cellular Automata to account for geographical interrelationships (Houet et al., 2016; Marshall and Randhir, 2008). A very interesting technique has been to combine the top-down (demand-driven) and bottom-up (local conversion) processes of land cover change by proceeding to a simplification of local processes (van Asselen and Verburg, 2013; Verburg and Overmars, 2009). Despite a huge bibliography both in climate change and land cover change, scenario analysis over the past 25 years has mostly focused on climate change projections, while the impact on land use and land cover has been neglected. Titeux et al. (2016) found that only 11% of the 2313 studies analyzed have included both land cover and climate changes. Also based on a large review, March et al. (2012) have called this a "hegemony" of climate as a driver of change. Furthermore, the implementation of land cover change techniques appear to be tedious and does not account for the intensification of cropping patterns. The motivation of the present work is to take into account both land cover change and crop intensification in future scenarios of irrigation water demand, taking into account the impact of climate change.

The hypothesis is that the dynamics of land-use change and intensity of the cropping patterns (hence irrigation water demand) can be reduced to a synthetic monthly $K_c$ time series for every irrigated area of the studied region. Each synthetic time series would then account for the spatial variability of cropping patterns inside each irrigated area. Unless no sudden change occurs, a statistical model accounting for the monthly trend of $K_c$, the impact of rainfall, and the effect of saturating the land cover with tree crops should give an accurate fit that would allow extrapolating into the next few decades. As the synthesis of Kc for separated irrigated areas would also decrease substantially the amount of information compared to a land cover approach, so that some information, like the amount of tree crops, should be retrieved back from the time series. This approach would prevent the need for tedious land cover classifications and reduce the difficulty of working with discrete values for developing future scenarios.

The objective of this study is to produce two scenarios of irrigation water demand for the different irrigated areas of the Tensift watershed in Morocco. One is the trend, the other one is an alternative scenario derived from a narrative scenario. To do so, a method for simulating and extrapolating $K_c$ is proposed and is tested for the 2000-2016 period. The future time series of $K_c$ is used with climate change scenarios to obtain an estimate of future irrigation demands. As the strategy of irrigation for tree crops is treated very differently from annual crops in the Tensift watershed, the yearly percentage of tree crops is also retrieved from the $K_c$ time series.

The content of the article is separated into four parts and a conclusion. The first part describes the study area and the dataset. The methodology is then detailed in three parts: 1) $K_c$ time series are obtained from remote sensing observations for the observation period, 2) a model is constructed to project the IWR based on simulating the evolution of $K_c$ and integrating climate change; and 3) an alternative scenario is built from the trend to reflect a narrative scenario built by local water managers. In the results section, the performance of the model and its ability to make projections are analyzed. The alternative scenario is compared to the simulated trend. In the discussion, other paths are discussed, in particular those which are related to irrigation management.

## 2 Study Site and Data

### 2.1 Study Area: the Haouz Plain in Morocco

The study area includes the Haouz Plain, underlain by a large Plio-Quaternary alluvial aquifer. It is located within the Tensift watershed in southern Morocco around the city of Marrakech (Figure 1). The plain is bordered at the south by the High Atlas range. The High-Atlassian watersheds that peak at 4167 masl receive about 600 mm/year of precipitation of which a major portion falls as snow; 25% of the streamflow is generated by snowmelt (Boudhar et al., 2009). The Plain, which is located at altitudes ranging between 480 and 600 masl, has a semi-arid climate (rainfall of 250 mm/year and potential evapotranspiration of 1600 mm/year) with mild, wet winters and very hot and dry summers. Irrigation is essential for good crop development and yield. The so-called "traditional" areas (purple) which originally received water from the Atlassian wadis are more and more dependent on groundwater. The areas irrigated from lake reservoirs (green) receive water from the

Lalla Takerkoust reservoir, the Moulay Youssef reservoir and also from the Hassan 1er reservoirs through a 140km canal
called the Rocade Canal. The rest of the irrigation water comes from the groundwater.

## 2.2 Data

In order to compute the crop-water demand, Regional Climate Model (RCM) simulations over the Mediterranean region
from the Med-CORDEX initiative (https://www.medcordex.eu, Ruti et al., 2016) were utilized to provide future values of
near-surface atmospheric variables, namely the 2m air temperature and total precipitation. RCM results from two contrasting
greenhouse-gas concentration RCP (Representative Concentration Pathway) trajectory scenarios were selected: the RCP4.5
scenario, which represents the optimistic scenario for $CO_2$ emissions (a stabilization of emissions) and the RCP8.5
(increasing emissions). Simulations from Centro Euro-Mediterraneo sui Cambiamenti Climatici (CMCC), Météo-France
(CNRM) and IPSL-Laboratoire de Météorologie Dynamique (LMD) were used in the current study. The three RCMs give a
good representation of the spread typically found in such simulations. The so-called "delta-change" or perturbation method
(Anandhi et al., 2011) was used to downscale the RCM data using the Global Soil Wetness Project Phase 3 forcings
(GSWP3: http://hydro.iis.u-tokyo.ac.jp/GSWP3) as the reference historical data. Statistical scores (bias and daily correlation
coefficients) were computed between the GSWP3 product and 11 SYNOP stations over north-west Africa over the 1981-
2010 period. GSWP3 well represents the variability of temperature relative humidity and total precipitation. The correlation
coefficients computed between both monthly time series are 0.99, 0.87, and 0.74 for temperature, relative humidity, and
precipitation, respectively. Standard deviations are quite small for the three variables (0.01, 0.03, and 0.12 respectively). At
the daily time step, GSWP3 well represents the day-to-day temperature variability and especially the relative humidity.
However, GSWP3 has more errors in terms of the daily evolution of precipitation (r=0.08, stdev=14.6). In semi-arid regions,
where precipitation is infrequent and often convective, it is difficult to reproduce the precipitation events at the actual time.
In terms of bias, the absolute temperature and relative humidity values are close to that observed: 0.22 °C and 0.54%
respectively, meanwhile precipitation is underestimated by approximately 10% on average, but this bias is not always
consistent from one station to another. The 0.5° data were resampled to a resolution of 1 km, applying a correction of altitude
between the GSWP3 geopotential and a digital elevation model (GTOPO30 available at https://www.usgs.gov) for air and
dew point temperatures.

The resulting downscaled and rescaled (to 1 km resolution) dataset of meteorological variables (Rain, Temperature, Wind
Speed, Relative Humidity and Global Radiation) extends from 2000 to 2050.

For generating vegetation time series, we used the temporal the product MOD13Q1 Collection 6 (K. Didan, 2015) wich
provide 16-day composite series of Normalized Difference Vegetation Index (NDVI) from the Moderate Resolution Imaging
Spectraradiometer (MODIS) at a spatial resolution of 250 m. This product is computed from atmosphere-corrected (Justice
et al., 2002), daily bidirectional surface reflectance observations, using a compositing technique based on product quality

(Wan et al., 2015). We used the 16-day composites to reduce cloud coverage. The data has been compiled for the period 2000-2016.

## 3. Methodology

### 3.1 $K_c$ assessment from Remote Sensing Time Series

Neale et al. (1990) proposed to estimate the $K_c$ coefficient from vegetation indices obtained from remote-sensing imagery. In this work, we have been using empirical linear equations where the slope (a) and intercept (b) have been previously calibrated for common crop type in local field experiments (Duchemin et al., 2006; Er-Raki et al., 2007, 2008):

$$K_{c_{sat}} = a * NDVI + b \qquad (2)$$

Equation (2) accounts for combined Evaporation from the soil and Transpiration from the crop in a very simple way, but as no water balance is performed, several assumptions must be kept in mind: 1) the extra evaporation due to wetting events
(rainfall, irrigation) is not computed, but is assumed to be represented in the calibrated NDVI/$Kc_{sat}$ relationship, which is related to the precipitation of the year used for calibration. The inability of this method to increase the evaporation part of evapotranspiration when frequent small rainfall events occur should be assessed and corrected for, either at the level of the $ET_c$ computation or in the $P_e$ computation; 2) the lower evaporation due to a reduction of the wetted fraction characteristic of micro-irrigation is not taken into account; 3) The crop water stress is not taken into account in this calculation, as the time
step of one month causes this to be quite complex to represent. However, significant stress may impact the evolution of the NDVI signal. The approach of estimating $Kc_{sat}$ is displayed in the light-gray area of Figure 2.

### 3.2 Trend projection of $K_c$

The simulation of $K_c$ was carried out as follows: A linear adjustment from the original $Kc_{sat}$ is obtained from remote sensing, secondly, this first guess is corrected according to the rainfall of the year, finally a correction is made according to the
percentage of tree crops. Figure 3 illustrates the quantities used to perform this calculation. The four steps detailed below were reproduced for each of the 29 irrigation areas of the study.

(1) The first step was to perform a linear adjustment (eq. 3). For each month $M$, of the year $y$, a linear least-squares fit was made to the $Kc_{sat}$ values for all years in the series, thus we obtained a set of 12 monthly regression equations for $Kc_{sat}$ as a
function of the year. These twelve linear regressions thus formed the time series $Kc_{lin}$. An example is plotted in blue in Fig. 3. :

$$Kc_{lin_M}(y) = a1_M.y + b1_M \quad (a1 \ and \ b1 \ are \ fitted \ to \ Kc_{sat_M}(y) \ for \ M \in [1,12]) \qquad (3)$$

(2) The second step consisted of correcting the first approximation of $Kc_{lin}$ for the impact of rainfall. In this semi-arid region, sowing and the development of the vegetation depend very much on the time distribution and accumulation of rainfall during the agricultural season. We chose March as the month most representative of these interannual differences (Le Page and Zribi, 2019). On the one hand, we calculated the difference in $K_c$ ($\Delta K_c$) between the observation ($Kc_{sat}$) and the linear regression ($Kc_{lin}$) in the month of March as in Eq. (4). On the other hand, the rainfall accumulation $P_y$ was summed between September and March according to Eq. (5). This has been done for each year from 2000 to 2016. We found that the relationship between $\Delta K_c$ and $P_y$ was best modeled using a second-degree polynomial according to Eq. (6). An example of this fit is given on the right side of Fig.3.

$$\Delta Kc(y) = Kc_{lin}(y) - Kc_{sat}(y) \tag{4}$$

$$P_y = \sum_{m=sep}^{mar} P_m \tag{5}$$

$$Kc_{cor}(P_y) = a2. P_y{}^2 + b2. P_y + c2 \quad (a2, b2, c2 \text{ are fitted to } \Delta Kc_M \text{ for } P_y) \tag{6}$$

(3) In the third stage, the tree crops and their preservation were taken into account. In fact, in semi-arid Mediterranean regions, evergreen tree orchards (olive, citrus) are the major crops responsible for green vegetation in the dry season of summer, although the cultivation of summer vegetables is also possible. In the Tensift region, orchards come first in order of preference for water allocation in agriculture, especially in dry years. The aim is to preserve the tree crops from drought. To ensure that the tree crop area would not be reduced from one year to the next, the minimum $Kc_{sim}$ was forced to not run in the opposite direction to the trend slope (it won't be reduced when the trend is growing and inversely). In other words, the yearly rate of minimum $K_c$ is growing, a new value is ensured to be greater or equal to the minimum value of the previous year. Finally, to ensure that the resulting synthetic $K_c$ would be in line with the proportion of trees in the irrigated area, the percentage of tree crop has been obtained during the dry season when it is assumed that there is no other crop than trees:

$$\%Trees(y) = \frac{minKc(y)}{Kc_{Trees}} \tag{7}$$

where minKc is the minimum $Kc_{sim}$ for the hydrological season (see the blue line on Fig. 3) and $Kc_{Trees}$ is the maximum $K_c$ for the tree crop considered (see the orange line on Fig. 3). For the study area, in which olive trees are the dominant tree crop, $Kc_{Trees}$ has been set to 0.55 (Allen et al., 1998). The potential maximum $K_c$ of each year (maxKc) was computed with the weighted sum as follows:

$$maxKc(y) = Kc_{max}. (1 - \%Trees(y)) + Kc_{Trees}. \%Trees(y) \tag{8}$$

where the non-tree crop area has a maximum $K_c$ of $Kc_{max}$, which was set according to winter wheat as in Allen et al. (1998). Winter wheat is a dominant crop of the study area, and its maximum $K_c$ (1.15) is among the highest known.

(4) The final estimate of monthly $K_c$ ($Kc_{sim}$) integrated the linear trend, the yearly variation due to rainfall and the limitation due to the coverage of tree crops (eq. 9).

$$Kc_{sim_M}(y, P) = \max\left[\min\left(Kc_{lin_M}(y) + Kc_{cor}(P), maxKc(y)\right), Kc_{min}\right]$$ (9)

where the variability of $K_c$ due to yearly rainfall ($Kc_{cor}$) was added to the first guess $Kc_{lin}$. It was ensured that this sum did not exceed the potential maximum $K_c$ ($maxKc$, Eq. 8). It was also ensured that it did not go below the $K_c$ of bare soil ($Kc_{min}$ = 0).

The set of equations 3 to 9 together model the future trend of $K_c$. The water demand was then computed from Equation 1, where $ET_0$ incorporated the weather conditions, and in particular the modifications of temperature predicted by the RCPs..

The performance of the model has been assessed using different sampling techniques to obtain calibration and validation data sets. First, calibration years are selected to be equally spaced along the time axis: one out of every two (8 years), one out of every three (6 years) and one out of every four (4 years) of calibration. For the 'every two years' sampling, there were two sets of calibration years, for the 'every three years' sampling, there were three calibrations sets, and so on. The three different versions of the model were run for the different combinations of calibration and validation data sets. The versions

tested were the linear fit ($Kc_{lin}$), the linear fit with rain correction ($Kc_{cor}$) and then with tree correction ($Kc_{sim}$). The performance indicators were averaged for each group (1/2, 1/3, 1/4). The years have also been separated into two groups: years with less precipitation are called the 'dry years' (2001, 02, 05, 07, 08, 11, 12, 14, 16) and the rest being called 'wet' years. Three statistical metrics have been computed: the determination coefficient $r^2$ (Eq. 10), the root mean squared error RMSE (Eq. 11), and the standard deviation of error stderr (Eq. 12):

$$r^2 = \sqrt{\frac{Cov(X,Y)}{\sigma X \, \sigma Y}}$$ (10)

$$RMSE = \sqrt{\frac{\sum_1^n (X - Y)^2}{n}}$$ (11)

$$stderr = \sqrt{\frac{\sum_1^n (X - Y)}{n}}$$ (12)

**3.3 Alternative evolution of $K_c$**

The set of equations 3 to 9 were used as a basis for simulating the long-term trend of $K_c$. The coefficients a1 and b1 of Eq. 3 were fitted with the data from the entire time series. The parameters a2, b2 and c2 of Eq. 6 were fitted using the rainfall data taken from the downscaled climate scenario.

The system of equations allows the consideration of various possibilities for "bending the curve" (Raskin et al., 1998), which

here means modifying the trend of the cropping scenario: (1) the parameters a1 and b1 in Eq. 3 can be modified to account for an overall change in the trend; (2) the coefficients a2, b2, and c2 in Eq. 6 can be modified to account for the influence of

cumulative rainfall. (3) The hypothetical law that determines that the rate of change in tree coverage always has the same sign could be relaxed so that a decrease of tree coverage could be possible.

As the status of the Haouz aquifer is becoming critical, alternative trend scenarios have been proposed by the Tensift Basin
Agency under the AGIRE Convention developed with the support of GIZ (*Gesellschaft für Internationale Zusammenarbeit)* (ABHT GROUP AG - RESING, 2017). The main drivers defining these changes include 1) an expansion of irrigated areas through groundwater pumping; 2) the conversion of surface irrigation to drip irrigation; 3) an intensification of arboriculture; and 4) an abandonment of irrigated cereals for vegetable crops. However, neither the plans nor the interviews specify the intensity or locations of these changes. Here, we do not address all of the measures considered, only those concerning land-
cover changes. A reduction in the rate of increase of olive trees areas by 50% is expected to begin in 2021. At the same time an intensification due to intercropping and the replacement of traditional Moroccan Picholine (100 to 200 trees per hectare) by spanish Arbequina with a density of 1000 trees per hectares wille take place. This is correlated with the expected production increase of 27%. The fate of citrus trees is less clear. The recent trend is a strong expansion, however the plan expect a stabilization of citrus orchards due to the lack of water especially in the western part of the basin. Finally,
aprogressive reduction of cereal crops (-722 ha/year) is expected. Here, we proposed to only modify the main trend of the $K_c$ curve in order to represent the idea of a pause in crop-cover expansion. The set of equations to simulate the bending of the curve are:

$$a1' = a1 . b_c$$
$$b1' = (a1 . b_c + b1) - y_c . a1' \qquad (13)$$
$$Kc'_{sim} = a1' . Kc_{sim} + b1'$$

A new slope (a1') and intercept (b1') were computed according to a bending coefficient ($b_c$) and applied for the desired years were $y_c$ is the beginning year of the bending. The bending coefficient was expressed as a percentage.
In order to demonstrate the feasibility of applying scenarios with this model, it was determined that the introduction of two bending points to the trend scenario (Eq. 13) gave a good representation of the vision of the narrative scenario (-50% on the trend in 2020 and another -50% by 2040). Note that the narrative scenario didn't give a detailed view of the changes so that those coefficients have been applied uniformly to the 29 irrigated areas.

## 4 Results

### 4.1 Performance of the simulated $Kc_{sim}$

Figure 4 shows the three statistical metrics of the fit for different calibration groups. The calbration is computed against a subset of the 16 years, where pne half (1/2) means 8 calibration years for 8 validation years, one third (1/3) means 5 calibration years for 11 validation years and one quarter (1/4) means 4 calibration years for 12 validation years. r2 were generally located around 0.5 which indicates an average fit, but it decreased when the calibration set size decreased. RMSE
ranged between 0.1 for half of the calibration years to 0.17 for the corrected methods for only four calibration years. It must

be noted that the Kcsim and Kccor versions lower the performances regarding RMSE when only 1/3 or 1/4 of the years are used. This is also true for the dry calibration years which is explained by the fact that the rainfall correction (Kccor) needs a good representativeness of extreme cumulative precipitations.

The trajectory of each irrigated sector can be very different, according to its mix of crop, cropping intensity, and evolution. Figure 5 shows an example of this diversity with four contrasted irrigated areas. The figure also shows the calibrated and original time series of $K_c$ when the calibration is carried out over the entire time series. The Chichaoua Private area gave the worst result. As can be seen, the low $r^2$ was mainly due to the poor distribution of the data at very low $K_c$. It is very likely that most of the area was not irrigated continuously during the whole study period. The N'Fis-RD is located on the outskirts

of the city of Marrakech, under high urban pressure. The negative trend was well predicted and the correlation score was satisfactory ($r^2$=0.77) despite the probably erroneous pause in the original $K_c$ data in 2006 and 2007. The Ourika traditional area is located at the outlet of the Ourika River. The scores were average ($r^2$ = 0.646, Slope = 0.747). The general growing trend was reproduced, especially for the base of the curve. The maximum and minimum $K_c$ simulated with the amount of annual rainfall was generally accurate, except for the two years 2002 and 2003. Those are extremely dry years where rainfed

cereals couldn't develop and there were strong irrigation restrictions. The Bouidda district is an area irrigated by the Moulay Youssef reservoir in the eastern part of the study area. It obtained one of the best scores ($r^2$=0.863). The area is dominated by cereals, which explains the typical seasonal $K_c$ curve. The increase in permanent tree crops was well simulated at the baseline. There was also an intensification of cereal crops that increased the peaks at the end of the study period. The lower $K_c$ years 2006 and 2007 were well simulated. The very dry year 2001 was simulated but did not reach the minima observed

in the original curve.

The performance of the algorithm may be summarized as follows. The simple linear relationship provided an average approximation. The rainfall correction improved the average performance if the calibration years were representative of the validation years, elsewhere it worsened the prediction. There was no visible effect of the tree correction during the period of

290 time where observations are available. The bounding constraint was basically not reached. Finally, the model results were much better if the calibration years were representative of the rainfall variability (wet/dry years). Given that the remote sensing dataset was processed for the 2000-2016 period (16 years), the performance of the prediction at the 2050 horizon should be comparable to the 1/3 test: $r^2$ = 0.5, RMSE = [0.1-0.14], stderr = [0.02-0.03].

## 4.2 Evaluation of the trend and alternative evolution under climate change

Figure 6 shows examples of the extrapolation of $K_c$ for four contrasted irrigated areas, with the trend and alternative $K_c$ scenarios and RCP4.5 and RCP8.5 climatic scenarios. The region of private irrigation in Mejjat was all but covered with drylands until the early 90s. A boom of groundwater exploitation has provoked the development of irrigated areas. $K_c$ spikes in the rare wet years, but most of the time, the amplitude of the signal is very small, showing that the area is mostly

composed of bare soils and tree crops. The trend projection predicts that the $K_c$ will tend toward an average of 0.2 by the year 2050, which means the impact of irrigation will be much greater than at the beginning of the XXI century. The alternative $K_c$ greatly reduced the trend and stabilized around 0.15. Ourika is an irrigated area located at the piedmont of the Atlas mountains. This area has traditionally relied on spate irrigation from the Atlas wadis fed by snowmelt. This area is also located over the Haouz aquifer where it benefits from recharge from below-river flows. The full coverage with tree crops was attained around 2040 in the Ourika area. Therefore, the synthetic $K_c$ of this area is controlled by the maximum $K_c$ value of equation 8, which was set as to the Kc of olive trees (0.55) which is the dominant tree crop of this region.

On the alternative trajectory, $K_c$ is less saturated, meaning that there is still a place for annual crops. As stated earlier, the N'Fis right bank irrigated area, which is located near Marrakech already had a trend toward the disappearance of irrigated crops. The two projections maintained the $K_c$ level near zero except for wet years. The Bouidda irrigated area in the Tessaout region follows a trend of reconversion toward tree crops without a trend of cropped area development, which is almost fulfilled by 2050. In the alternative scenario, this trend was controlled and the area maintained a mix of annual and tree crops.

In those four examples, the climatic scenarios of rainfall only had a minimum impact on the $K_c$ time series.

Figure 7 shows the relative evolution of water demand compared to the year 2000, and the expected impact of the "bending" effect for the four different planning areas in the watershed (see Fig. 1). The expected temperature increase between RCP4.5 and RCP8.5 is about 1°C by 2050, leading to a 2% difference in yearly $ET_0$. The impact on irrigation water demand is therefore minimal by the end of the simulation and has not been plotted in figure 7. However, we must keep in mind that if climate change only slightly impacts the water demand, it will probably reduce the water offer, because the run-off decrease will reduce the surface water available for irrigation (snowmelt, runoff, reservoirs). The Area Equipped for Irrigation (AEI) also remains unchanged while the actual irrigated area may change inside those AEIs. Those simulations showed that each area had it's dynamic (bearing in mind that sub-areas also had their own dynamics). The Mejjat area, which was mostly unexploited in the year 2000, has been experiencing significant growth since. We can also see that by 2040, the demands tend to stabilize in the trend lines owing to the saturation effect of the irrigated areas. In the alternative scenario, the reduction in the agricultural development rate stabilizes at a lower level. The N'Fis area showed the lowest increase, and the modeled trend towards a flat line showed that the area will be fully dominated by tree crops by 2050. The Haouz and Tessaout areas showed the most significant increases in irrigation water demand, as it almost tripled during the 50 years of simulation. However, the alternative scenario mitigated these increases.

## 5. Discussion

The proposed method allowed projecting irrigation water demand by including both the anthropogenic and climatic vector of changes. The output is given for the next thirty years for each demand site at the monthly time step which are scales commonly used by water planners. Various improvements to the method have been identified: 1) $K_c$ was determined by a pre-calibrated linear relation to the NDVI time series. If an actual evapotranspiration product (see for example (Xu et al.,

2019) would be available for the whole period of study and at a resolution compatible with the size of the irrigated areas, $K_c$ could be determined by computing the ratio between actual evapotranspiration products and reference evapotranspiration. 2) The gridded data were synthesized at the monthly time step for each irrigated area, by computing arithmetic averages, which resulted in a loss of information. If needed, other statistical indicators, such as the dominant land cover of the irrigated area could be kept for further analysis. 3) The main temporal trend was fitted with monthly linear regressions (Eq. 3). The degree of the regression could be easily adapted if the model does not fit adequately with the observations.

The very low difference in impact between the two RCP scenarios is interesting. It can be explained by the short time range (2050) which resulted in a difference of only 1°C and almost no difference in precipitation over the study region. It should be noted that the predicted impact of climate change on precipitation is more variable in space and less certain than that for temperature. It also may be due to the absence of the impact of including increasing atmospheric $CO_2$ concentration prescribed by the RCP scenario in our approach. Different strategies are possible. For example, Fares et al. (2015) proposed a modification of the ratio bulk resistance over stomatal resistance $r_s/r_a$ in the $ET_0$ equation based upon the $CO_2$ emission scenario. The low impact of RCP scenarios could also be explained by the fact that the potential decrease in the annual crop growth cycle duration associated with temperature rise is not represented. According to (Bouras et al., 2019), the reduction of the cycle can reach 30% at mid-century under RCP8.5 scenario for wheat in the study region. Similar results were obtained with the ISBA A-gs model (Calvet et al., 1998) when studying the impact of climate change on Mediterranean crops (Garrigues et al., 2015). It is also interesting to note that if the water demand for seasonal crops is reduced with the crop cycle duration, the yield may also be affected by extreme temperatures (Hatfield and Prueger, 2015) so that the trend might be affected in order to reach the production objective.

Finally, the assessment of irrigation water demand should also take into account different losses at the system level (storage, transportation and operating losses) and at the plot level (deep percolation and runoff). The future evolution of the irrigation framework toward more efficient systems such as drip could also be considered using equation 14. This equation of Gross Irrigation Water Requirement (IWR$_G$ expressed in m$^3$) is rewritten from equation 1:

$$IWR_G = 1/\sigma \frac{AEI}{10} \frac{\varphi ET_0 K_c - \alpha P}{\beta} \tag{14}$$

where AEI (hectares) is the area equipped for irrigation. The system efficiency $\sigma$ is the ratio of water delivered upstream to the water that is distributed to the plots (Blinda, 2012). The coefficient $\alpha$ produces the efficient rainfall similar to that proposed by (USDA-SCS, 1970). The coefficient $\varphi$ is used to account in a simple manner for a reduction of evapotranspiration due to micro-irrigation. $\beta$ is the effective irrigation coefficient; it is low for surface irrigation (50-70%) and close to 100% for drip irrigation. The behavior of equation 14 is displayed in Figure 8 for two sample years (September 2011 to August 2013) for the R3 irrigated sector where wheat is the dominant crop. According to those different efficiency configurations, IWR can vary from single to more than quadruple between the ideal best and the worst situation. However, note that those configurations are theoretical. The best case with drip is virtually unachievable in real conditions.

## 6. Conclusions

Owing to their inherent complexity, scenarios of agricultural evolution based on interviews are very difficult to translate into numbers (when and how much), and to represent spatially (where). The simple and flexible statistical approach proposed here is a possible solution for quantifying spatially distributed scenarios of agriculture evolution in the context of climate change and irrigated areas that are rapidly changing owing to socio-economical influences. This is the case for many Mediterranean areas like the Tensift watershed which was the case used for the current study. The performance of the model was acceptable in most irrigated areas, giving monthly correlation coefficients of $K_c$ of up to 0.92 but with significant differences between the irrigated areas. It was shown that the prediction performance of the model was a function of the length of the calibration period. With 16 years of data (2000-2016), the prediction could only be done for two other periods of 16 years (until 2050). In order to demonstrate the flexibility of the model for scenario building, a local scenario of water resource management was reinterpreted to build an alternative scenario upon the trend scenario. Two climate scenarios (RCP4.5 and RCP8.5) were introduced into the trend and alternative scenarios. The local anthropogenic scenarios of irrigation water demand had rapidly changing dynamics and were spatially contrasted. On the contrary, the difference of impact between the two climate change scenarios appeared to be very small over the next 30 years, although the impacts of atmospheric carbon concentration on stomatal controls of the plant were neglected. Finally, the discussion showed that the approach could easily be combined with irrigation efficiency scenarios. The conjunction of the present approach with irrigation efficiencies would probably be a suitable combination for the modeling of water resources management. The approach can be applied to other regions where there is a significant growth of irrigated areas. The MODIS products are available over all the globe so that the retrieval of a 20-year long time series of crop coefficients can be done anywhere. Different alternative techniques exist to retrieve Kc so that it is possible to switch between methods. A caveat of using this data is however its relatively low spatial resolution (250 m), which implies that some field-scale details might be missed or undersampled. The separation of the territory between different irrigated areas seems adequate in particular when using a water allocation model. The synthesizing algorithm is efficient and can be applied in different regions. The model itself is mainly linear, even if it is corrected by yearly precipitation and an eventual saturation due to the extension of tree crops. In other situations, for example, a late expansion of irrigated areas, or where a reduction of the expansion is already noticeable, it will be necessary to switch to a non-linear system. A simple ordinary least square adjustment has been used to fit the coefficients, but other more sophisticated techniques could be used. Those different questions are being taken into account in the frame of the AMETHYST project over the Merguellil watershed in Tunisia.

## Data Availability

Some or all data, models, or code generated or used during the study are available in a repository or online in accordance with funder data retention policies. (K. Didan 2015, https://www.medcordex.eu). Some or all data, models, or code used

during the study were provided by a third party (GSWP3). Some or all data, models, or code generated or used during the study are proprietary or confidential in nature and may only be provided with restrictions (ABHT GROUP AG - RESING, 2017).

## Code availability

Models or code generated or used during the study are available from the corresponding author by request.

## Author contribution

MLP analyzed and processed the satellite and meteorological data and proposed the model. LJ and AB supervised the work and reviewed the paper and took part in critical discussions. AB disaggregated the climate change forcings. YF participated in the formulation of the research question and the writing of the paper. BB coordinated the work with the Tensift Basin Agency. LJ, SK and MZ participated in the research of funding and reviewed the paper. All authors read and agreed with the 405 manuscript.

## Competing interests

The authors declare that they have no conflict of interest.

## Acknowledgments

We thank NASA for kindly providing us with the TERRA-MODIS NDVI products and Dr. H. Kim (U. Tokyo) for providing 410 the atmospheric forcing data from the Global Soil Wetness Project. We would also like to thank the MedCORDEX providers for making their Regional Climate Data available. We are especially grateful to the Tensift Basin Agency (ABHT) and the Haouz Agriculture Office (ORMVAH) for making their data available for the integrated modeling. Finally, we thank the AGIRE project for providing the tentative scenarios of the Tensift watershed.

## Financial support

This study was funded by the following projects: ANR AMETHYST (ANR-12-TMED-0006-01), CNRST SAGESSE (PPR/2015/48, program funded by the Moroccan Ministry of Higher Education), ERANET CHAAMS (ERANET3-062 CHAAMS) and the LMI TREMA project.

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

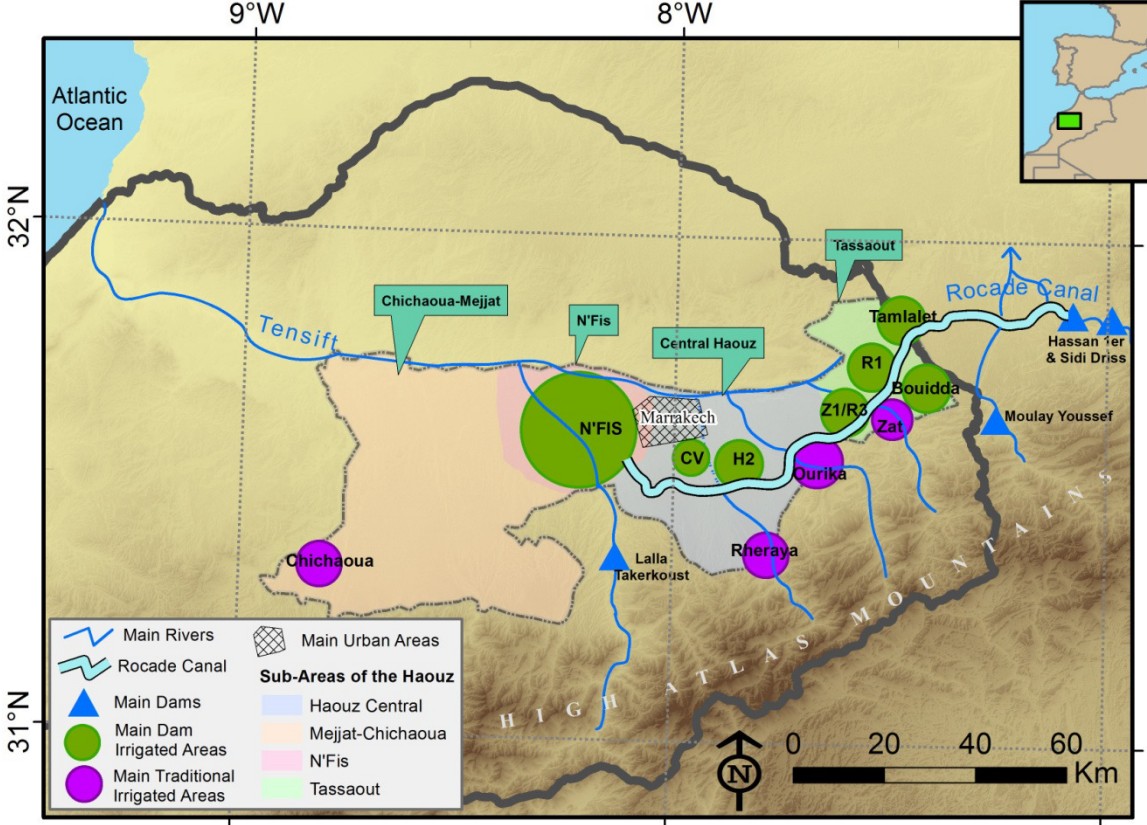

**Figure 1: The Haouz-Mejjat aquifer is located within the Tensift watershed (grey) and separated into four subareas. The most important traditional and dam irrigated areas are shown on the map. The N'Fis area is a mix of traditional and modern irrigation. All of these areas also use groundwater, but outside them, the irrigated areas (private irrigation) rely exclusively on groundwater.**

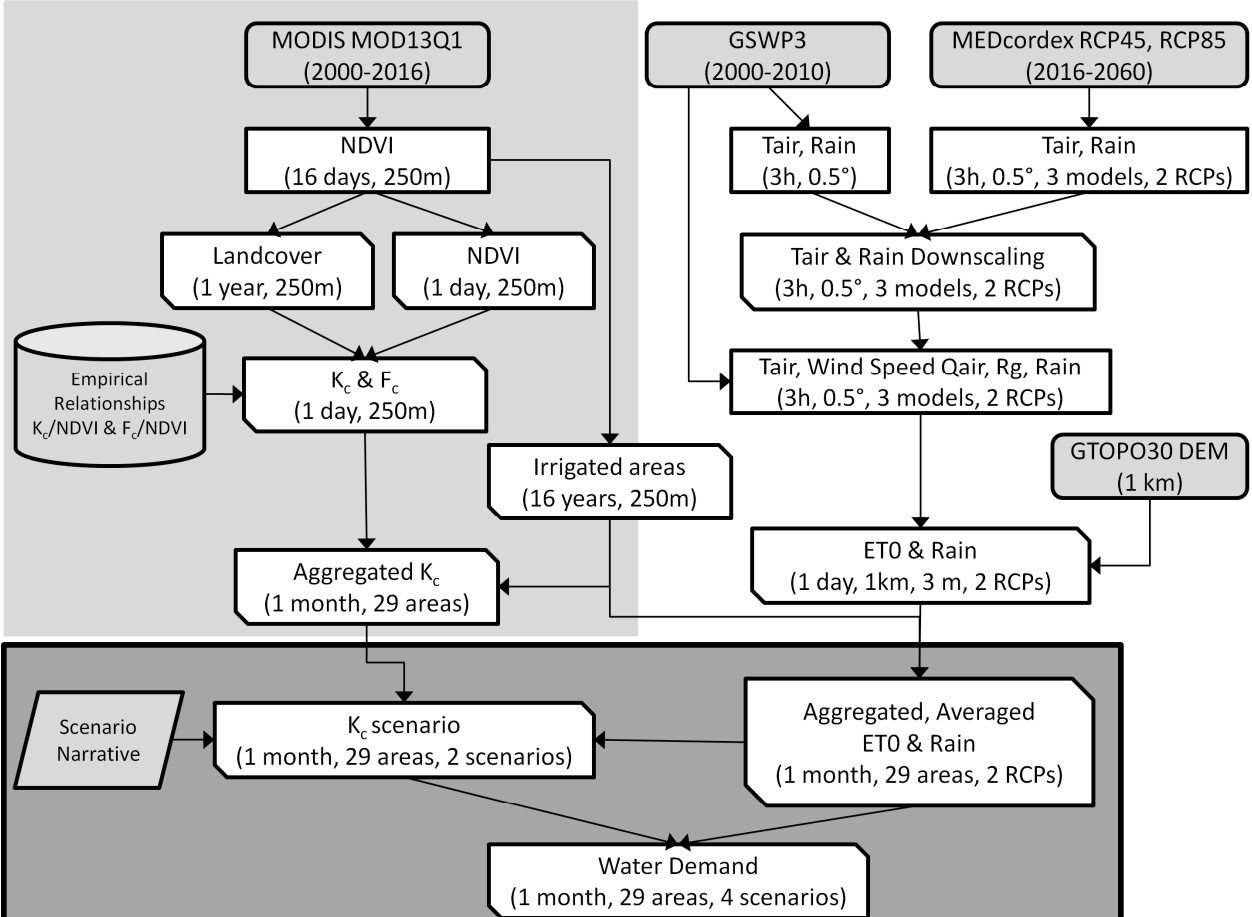

**Figure 2: Processing Flowchart. The light gray area of the Flowchart was presented in (Le Page et al., 2012). The calculations on $K_c$ are performed on 29 irrigated areas. The input data are indicated in gray boxes. Variable extraction is indicated by a square box. Processings (and their results) are indicated by boxes with rounded corners. The indications between parentheses show the temporal, spatial and scenario resolutions. The scenario generation described in this article is indicated in dark gray.**

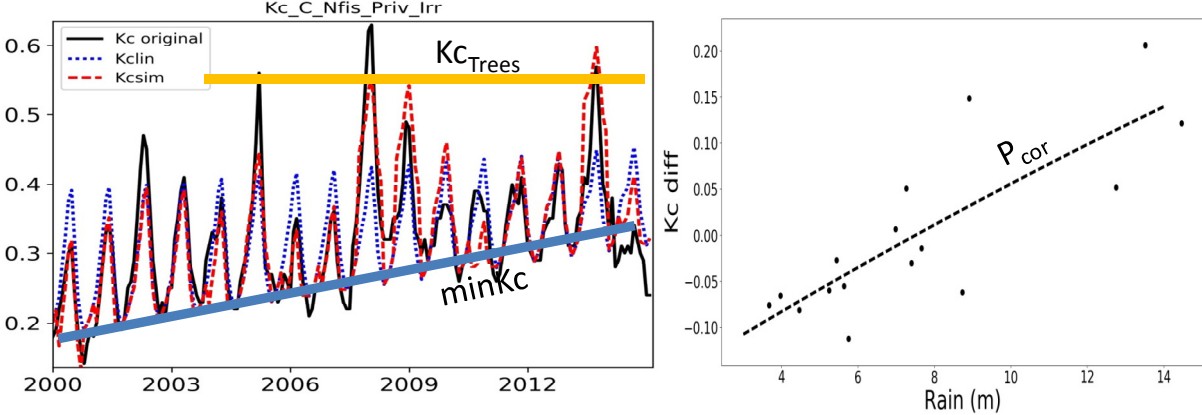

**Figure 3: Quantities used to perform the simulation. The example illustration on the left depicts the synthetic $K_c$ of the N'Fis private area where a mix of tree crops (olive, orange, apricot) and cereals crop (mainly wheat) are cultivated. The peak occurs during March/April when cereal crops are at their maximum development, the valleys correspond to the summer months when there are no seasonal crops. The three curves are the $K_c$ time series obtained from remote sensing processing (solid line), the linear estimation (Kclin, Eq. 3, dotted blue line) and the estimation corrected with rainfall ( Kcsim, Eq. 6, dashed red line). KcTrees has been set to 0.55 in Eq. 8, and minKc is the yearly minimum of Kc on Kcsim. On the right, the fitted correction curve of $K_c$ according to the cumulated rainfalls (Pcorr) is almost linear in this example.**

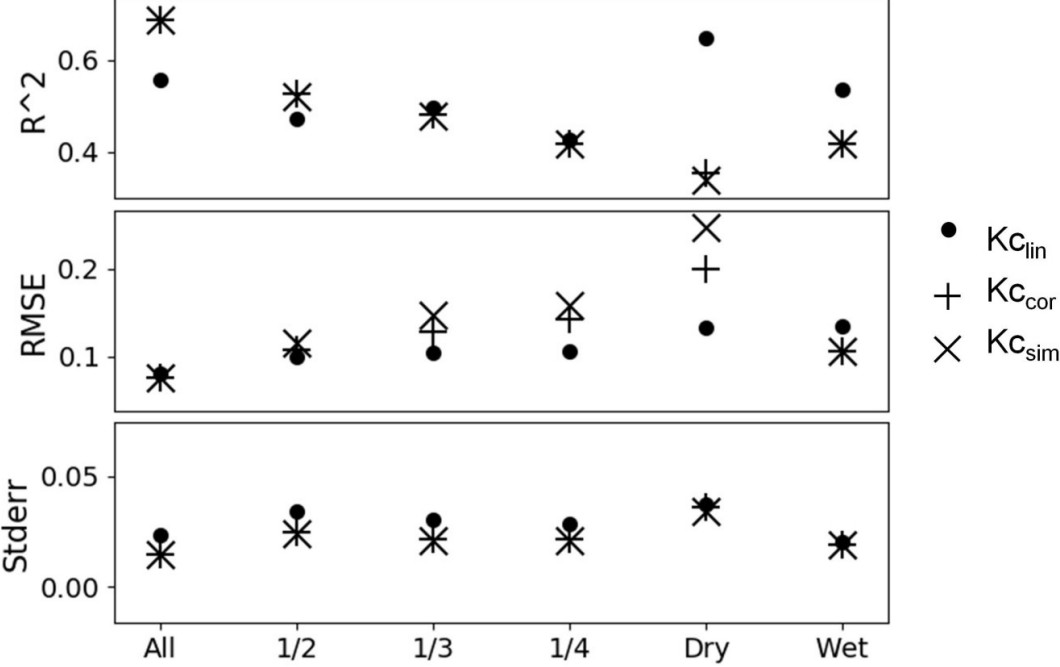

**Figure 4: Performance indicators ($r^2$, RMSE, stderr) for the three variations of the model ($Kc_{lin}$, $Kc_{cor}$, $Kc_{sim}$) as a function of the splitting strategy for determining the calibration/validation data set.**

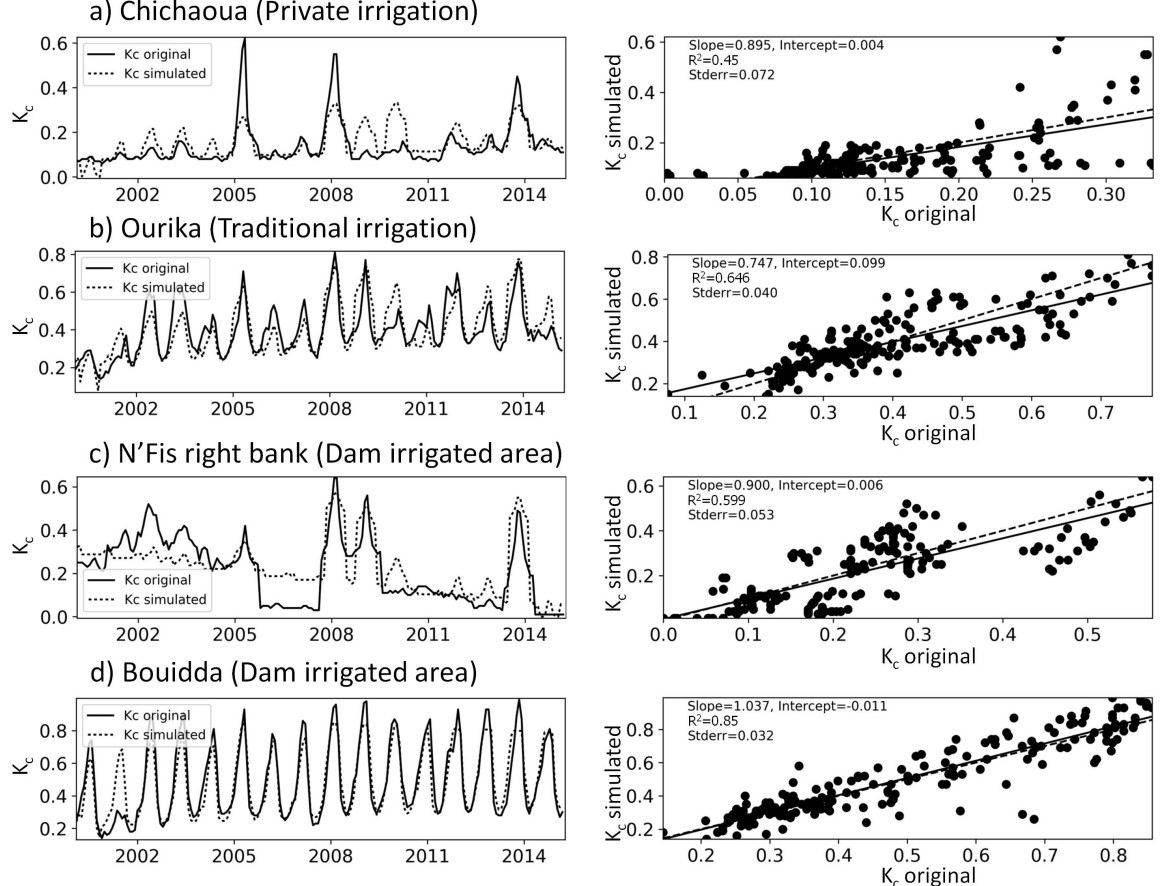

**Figure 5: Examples of $K_c$ simulations compared to the original time series for four selected areas.**

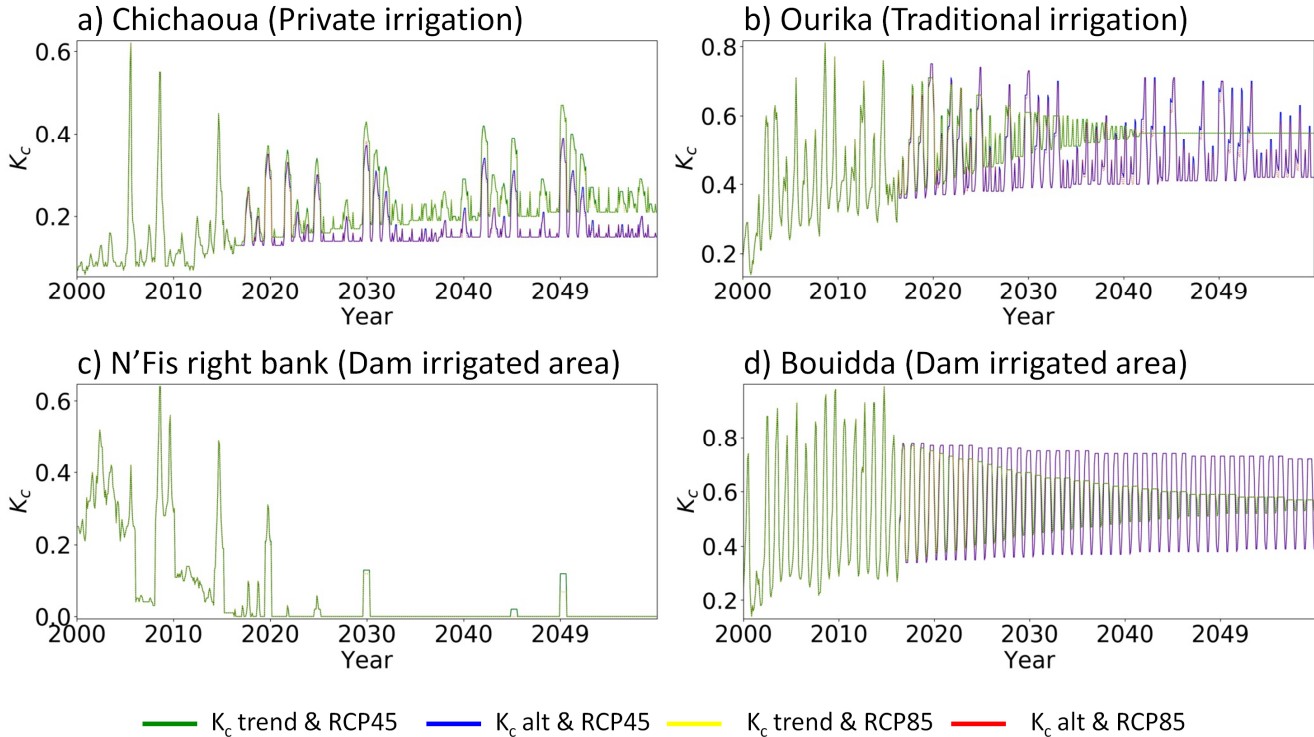

**Figure 6: Examples of long-term simulation of K$_c$ in four different irrigated areas with the trend and alternative scenarios of K$_c$ and the RCP4.5 and RCP8.5 climatic scenarios.**

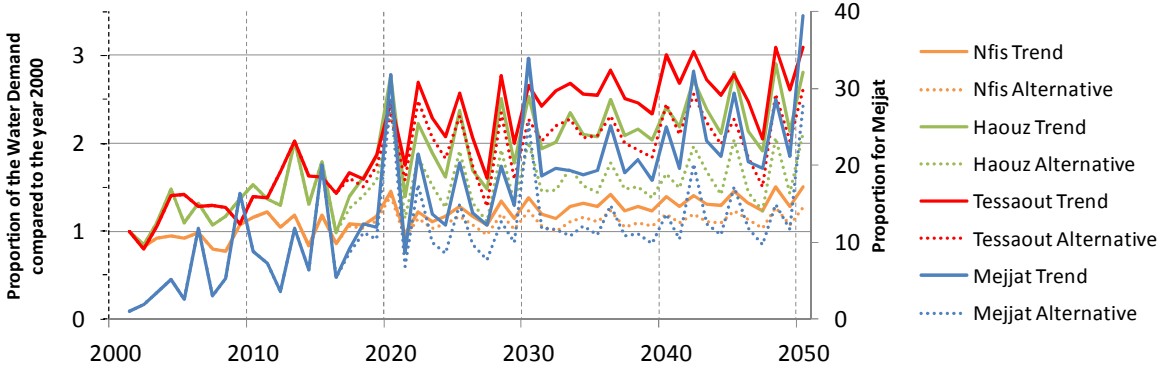

**Figure 7: Trend and alternative scenarios for irrigation-water demand in the four planning areas of the Tensift with RCP8.5.**

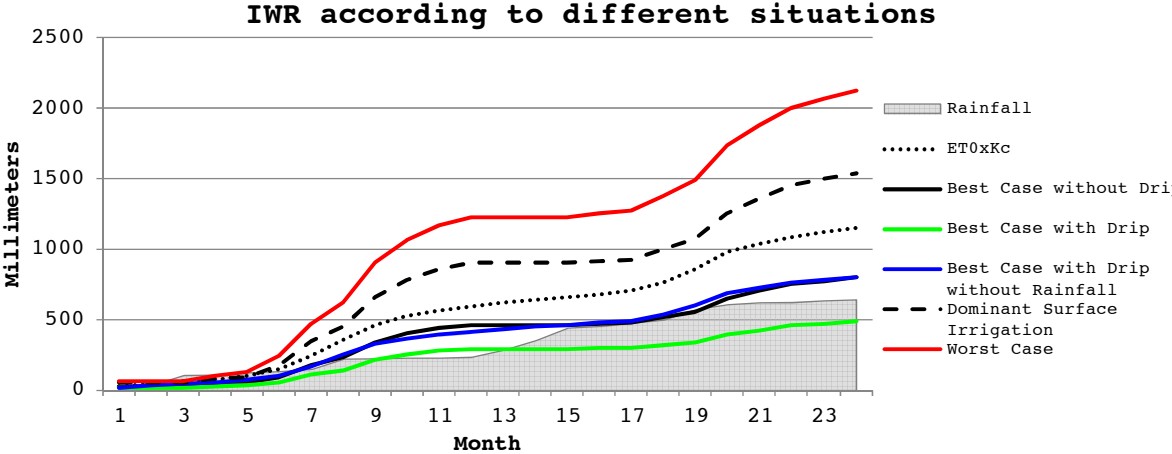

**Figure 8: IWR$_G$ according to different efficiencies for two years of example (September 2011 to August 2013) in the case of an irrigated wheat. In the best case without drip curve, all coefficients are set to one except the efficient rainfall which is set to a classical 0.75 (σ=1, φ=1, α=0.75, β=1). In the best case with drip, The σ coefficient of localized irrigation is set to 0.7 (σ=1, φ=0.7, α=0.75, β=1). The best case with drip but without taking into account rainfall simulates the irrigation practice done to prevent soil**

**leaching (σ=1, φ=0.7, α=0, β=1). The dominant surface irrigation simulates an irrigated sector where surface (gravity) irrigation dominates (α=0.5) and where the system performance is good (σ=0.9) (σ=0.9, φ=1, α=0.9, β=0.5). The worst-case depicts an irrigated sector with low irrigation efficiencies (σ=0.75, φ=1, α=0.75, β=0.5).**