# Peer review of "Projection of irrigation water demand based on the simulation of synthetic crop coefficients and climate change"

_Hydrology and Earth System Sciences, 2020_

## Referee Comment (RC1) · Anonymous Referee #1 · 12 Aug 2020

In this manuscript, synthetic crop coefficients were used to resolve irrigation demands in Southern Mediterranean area, Morocco. In general, the manuscript is well-written and I believe the proposed results could be useful to local engineers. However, I don't really see much contribution in terms of scientific novelty to the academic community. It is closer to an engineering practice with climate change scenarios. Therefore, I have to suggest reject from publication.
* * *

---

## Referee Comment (RC2) · Anonymous Referee #2 · 16 Sep 2020

This manuscript describes an enhanced approach to estimate crop coefficients, using corrected linear and tuned multilinear regression equations, to project future irrigated crop water demands, under two given climate scenarios for an irrigated region in Morocco.

Overall the paper is nicely written, and an adequate number of background studies and references are offered throughout the paper. The final results presented seem somewhat sufficient to support their approach and methods described. However, not many results are shown or addressed that support the title of the paper, and how the climate scenarios have any real impact on the irrigation water demand. The paper is

organized somewhat well, but some key details and background seem to be missing or making connections between each of the sections, especially in the Results and Discussions section. Some suggestions are offered below to hopefully help improve and further clarify the results of this paper. Finally, a couple of key conclusions are offered towards the end, however, one of the limitations of this study is that the approach was only applied for a small region in Morocco and not much discussion is given in how it may be applied to other regions, crop types, etc.

Major comments:

The introduction seems somewhat disconnected in how the authors present their case, hypothesis, background, and approaches. Also, more detailed connections should be drawn between each of the paragraphs to help with the flow and motivation more than simply introducing some key concepts.

Page 8, lines 225-230: This summary paragraph of the results related to Figure 4 refer to calibrated and validated years. However, in the previous paragraph in Section 4.1, the authors describe in the second sentence that the plots include calibration "over the entire time series". Then there is Figure 5, which is never mentioned or hardly discussed in the results discussion, though some of the statistical results are discussed, e.g., such as the last sentence on page 8 (lines 230-231), for the 1/3 test. Please make sure to address these issues in a future submission, and that results are well described and figures are discussed appropriately.

Minor comments:

Abstract: The abstract text seems to skip critical information when relating the future climate scenarios and the use of the derived synthetic crop coefficients. For example, on line 19, the authors mention "loss of performance . . . is to be expected for two time periods after the observations (2050)". How does "2050" relate here to the "two time periods"? What are they? Then the authors mention "This flexible system of equations" – what system of equations? The authors have room to increase the content of the abstract to help the summary make sense for readers potentially interested in reading the article. However, it is hard to follow the summary of results and methods introduced by the current way it is written. Also, what are the "two agricultural scenarios" mentioned on line 22? Again, it would be useful to at least provide some introduction or link back to early in the abstract the ties when such references are made.

Page 2 – Lines 33-34: Please clarify a bit more here what is defined by "dynamics of irrigated areas is strong". What dynamics are specifically referring to, e.g., overexploitation of groundwater, changing climatic conditions, etc.?

Page 3 – Lines 65-69: Please more fully describe and support with previous studies the types of approaches (i.e., "classic approach" vs. "a curve fitting" one) that are mentioned here. The language used here is somewhat vague on the background of the different approaches.

Page 3 – Line 79: Add either a comma, or more appropriately a semicolon, and the word "and" between "climate change" and "3) an alternative scenario..."

Page 3 – Line 86: The figure referenced here, "Figure 3", should be "Figure 1".

Page 3-4, Lines 94-102: This last part of the paragraph (Section 2.1) seems more appropriate in the Introduction section as part of the motivation in conducting this study. Authors may want to consider moving part or most of this description to the Introduction.

Page 4, Lines 114-115: Which station surface network data were used to evaluation the GSWP3 forcing fields? Or are you citing values from another published reference here? Please further specify the source for this GSWP3 validation and results.

Page 4, Line 117 (last sentence here): Which "meteorological dataset extends from 2000 to 2050", e.g., the downscaled RCP fields? Please further clarify this statement or move to a different location within this paragraph to accompany the appropriate data set description.

Page 4, Line 118: Please spell out the full acronyms for "MODIS" and "NDVI".

Page 5, Line 132: "small rainfalls" should simply be "small rainfall events . . .".

Page 7, Line 179: "GCMs" is referred to here, but nowhere in the paper is this mentioned. Did the authors mean to refer to the "RCMs" or "RCPs"?

Page 7, Lines 181-188: The authors indicate the years selected for calibration and validation, however, only the calibration years are noted here and how they are grouped. How are the validation years then selected separate from the calibration years? Please describe in more detail and clarify why the use of the different calibration year set groupings were selected.

Pages 7-8, lines 195-205: Please provide a bit more detail in how the upper limit of the crop-cover expansion is used to "bend the curve" and derivation of the coefficient, bc, is estimated.

Page 8, line 236: Why would Kc values "plummet in the rare wet years" and only for the Mejjat region? Wouldn't more rain simply mean less irrigated water demand but Kc values remain up? Please explain further why this occurs.

Page 8, line 238: "XX century" – do the authors mean at the beginning of the 21st Century, starting around 2000? Please check what was is intended here.

Page 9 + Figure 6: The results described here in relation to Figure 6 make many assumptions about how future crop coefficient values and irrigated scenarios play out under the different climatic scenarios and alternative Kc approach. The authors describe some of the results, but there appears to be some issue with how a value approaches certain constant values, such as for Ourika, that it makes it somewhat difficult to understand the results fully. The authors may want to revisit whether these results are actually robust and update their description here, as needed.

Page 9, line 240: Change "feed" to "fed".

[Figure]

Figure 2: Make sure "NDVI" is consistently used throughout the figure. For example, noticed "Ndvi" in the "Empirical Relationships" container should be capitalized. Also, in the caption, "parenthesis" should be changed to "parentheses".

Figure 3, title and caption: It would be helpful if details of which irrigated area, crop type, and month were highlighted in this example figure and caption.

Figure 4: Do these plots reflect the simulated Kc values of Equation 9? Authors may want to specify that here in the caption.

Figure 6: It is difficult to distinguish the four grey lines from each other in this plot and what is conveyed in the legend. It might be better to switch or replace some of the time series with colored lines.

Figure 8: "gravitary"? Did the authors intend to indicate "gravity" irrigation type? Authors might want to indicate that the graphic is showing "cumulative" IWR and rainfall for the two year period.

---

## Author Comment (AC1) · 15 Oct 2020

**In this manuscript, synthetic crop coefficients were used to resolve irrigation demands in Southern Mediterranean area, Morocco. In general, the manuscript is well-written and I believe the proposed results could be useful to local engineers. However, I don't really see much contribution in terms of scientific novelty to the academic community. It is closer to an engineering practice with climate change scenarios. Therefore, I have to suggest reject from publication.**

Dear reviewer,

first, thank you for the reading of our manuscript. Your two comments about the fact that the manuscript is well-written and could be useful for local engineers are welcome but we obviously disagree with the second part of your assessment. The rationales of the proposed method, that probably were not properly explained in the previous version of the manuscript's introduction are as follow:

- There are relatively few articles which take into account both the anthropogenic and the climatic trend for estimating irrigation demand in the future. Those are generally studied separately **(March et al., 2012a; Titeux et al., 2016)**.
- Most of the works are based on land cover change approaches and deterministic models of crop functioning or of plant water needs. **(**see for example **Fader et al., 2016; Malek et al., 2018)**. This means that those approaches are rather complex to implement, in particular for managers. They also need a large and representative data set of "training" data for the land cover change implementation. In addition, they are not suited to represent change of practices such as intensification.

We propose a simple semi-empirical method based on the observed agricultural trend that can be easily assessed from coarse-scale remote sensing time series freely available from MODIS for instance and, more recently from Sentinel-2 instruments. These trends can be modified in response to alternative scenarios that could be triggered by public policies for instance. To our knowledge, there are no similar empirical approaches based on observed time series in the literature and we believe this contribution could be relevant both to the academic community and local engineers. Of course, the strong limitation of the proposed approach is that the area of study must be subject to agricultural trends which is very likely in several agro-systems in the world facing conversion to cash crops or intensification.

In response to the reviewer's comment, the introduction has been significantly rewritten to detail the rationales of the proposed approach and its domain of application.

First, the approach is motivated by the important growth of irrigated areas both in terms of cropping density and cropped area in the Mediterranean zone and more specifically in the Tensift watershed in Morocco:

To satisfy the continuous increase in food demand associated with population growth, the agricultural sector has been asked to pursue its already initiated process of conversion towards agricultural intensification and above all towards a sharp increase in yields. This context goes hand in hand with the increase in food trade. The replacement of traditional crops by more financially attractive crops is already underway **(Jarlan et al., 2016)**. In the "growth" scenario (which is more or less the actual trend), presented by **(Malek et al., 2018)**, the annual production of cultivated land increases by 40% and, the production of permanent crops increases by 260%. In the "sustainable" scenario, annual crop production and tree production increase by 30% and 38%, respectively. As a consequence, irrigation water needs are expected to increase **(Fader et al., 2016)**. The expansion and intensification of tree crops will also further rigidify the demand for agricultural water and increase the pressure on groundwater reservoirs **(Jarlan et al., 2016)** in order to keep tree crops alive during drought events **(Le Page and Zribi, 2019; Tramblay et al., 2020)**. This study is carried on in the Tensift basin in Morocco, where the increase in the irrigated area and the intensification of irrigation during recent decades have caused a long-lasting drop in the groundwater table **(Boukhari et al., 2015)**. A multi-model analysis of the area **(Fakir et al., 2015)** has shown that the groundwater table falls from 1 to 3 m/year and that the mean annual groundwater deficit (about 100 hm$^3$ since 2000) is equivalent to 50% of the reserves lost during the previous 40 years. Among the main causes of this depletion, is a reduction and higher irregularity of precipitation **(Marchane et al., 2017)** for crop growth and groundwater recharge, a reduction of snow water storage **(Marchane et al., 2015)**, an increase and intensification of irrigated areas, and a progressive conversion to arboriculture due to national strategy. Since irrigation relies increasingly upon groundwater abstraction, questions are inevitably raised concerning the future of local agriculture and groundwater.

Secondly, the rationale for making this approach is now explained as follow:

There is a significant amount of literature about the estimation of land use and land cover changes **(Mallampalli et al., 2016; Noszczyk, 2018)**, with various techniques to estimate or predict them. Many land cover change approaches are based on the transition probability concept which was introduced by **Bell, (1974)**. They have been eventually connected to Cellular Automata to account for geographical interrelationships **(Houet et al., 2016; Marshall and Randhir, 2008)**. A very interesting technique consists of combining the top-down (demand-driven) and bottom-up (local conversion) processes of land cover change by proceeding to a simplification of local processes **(van Asselen and Verburg, 2013; Verburg and Overmars, 2009)**. Despite a large bibliography both in climate change and land cover change, scenario analysis over the past 25 years has mostly focused on climate change projections, while the impact on land use and land cover has been neglected to a large extent. **(Titeux et al., 2016)** found that only 11% of the 2313 studies analyzed have included both land cover and climate change. Also, based on a large review, March et al. (2012) have called this a "hegemony" of climate as a driver of change. Furthermore, the implementation of land cover change techniques appear to be tedious and does not account for the intensification of cropping patterns. The motivation of the present work is to take into account both land cover change and crop intensification in future scenarios of irrigation water demand while taking into account the impact of climate change.

In addition, its potential application to other agro-systems is also further discussed in the conclusion. This was also a comment of reviewer 2:

The approach is relatively simple and can be easily applied to other regions where there is a high growth of irrigated areas. The MODIS products are available over all the globe so that the retrieval of a 20 years long time series of crop coefficients can be done everywhere. Different alternative techniques exist to retrieve Kc so that it is possible to switch between methods. A caveat of using this data is however its relatively low spatial resolution(250 m), which might miss some details. The separation of the territory between different irrigated areas seems adequate in particular when moving to a model of water allocation. The synthesizing algorithm is efficient and can be easily applied in different regions. The model itself is mainly linear, even if it is corrected by yearly precipitation and an eventual saturation due to the extension of tree crops. In other situations, like for example a late expansion of irrigated areas, or where a reduction of the expansion is already noticeable, it will be necessary to switch to a non-linear system. A simple ordinary least square adjustment has been used to fit the

coefficients, other more advanced techniques could be used. Those different questions are being taken into account in the mark of the AMETHYST project aver the Merguellil watershed in Tunisia

---

## Author Comment (AC2) · 15 Oct 2020

Dear reviewer, thank you for taking the time to review this article. you will find our response to your review in the attached PDF. best regards
* * *

---

## Author Comment (AC3) · 15 Oct 2020

**1. This manuscript describes an enhanced approach to estimate crop coefficients, using corrected linear and tuned multilinear regression equations, to project future irrigated crop water demands, under two given climate scenarios for an irrigated region in Morocco. Overall the paper is nicely written, and an adequate number of background studies and references are offered throughout the paper. The final results presented seem somewhat sufficient to support their approach and methods described.**

We would like to take this opportunity to express our sincere gratitude for the effort and all constructive comments made in reviewing this manuscript. We have tried to consider all of your proposed comments to make a revised version to meet the publication requirements.

**2. However, not many results are shown or addressed that support the title of the paper, and how the climate scenarios have any real impact on the irrigation water demand.**

Agree. The manuscript has been significantly rewritten, more specifically to be in-line with the title as suggested by the reviewer. See comments n°4 and n°5

**3. The paper is organized somewhat well, but some key details and background seem to be missing or making connections between each of the sections, especially in the Results and Discussions section. Some suggestions are offered below to hopefully help improve and further clarify the results of this paper.**

Agree. Thanks for your suggestions that have been taken into account, especially in the introduction (see comment n°4), the results (n° 6), and the discussion (n°4)

**4. Finally, a couple of key conclusions are offered towards the end, however, one of the limitations of this study is that the approach was only applied for a small region in Morocco and not much discussion is given in how it may be applied to other regions, crop types, etc.**

Agree. A section dedicated to the discussion on the genericity of the approach was added in the conclusion of the new version of the manuscript as follows:

The approach is relatively simple and can be easily applied to other regions where there is a significant growth of irrigated areas. The MODIS products are available over all the globe so that the retrieval of a 20-year long time series of crop coefficients can be done anywhere. Different alternative techniques exist to retrieve Kc so that it is possible to switch between methods. A caveat of using this data is however its relatively low spatial resolution(250 m), which implies that some field-scale details might be missed or undersampled. The separation of the territory between different irrigated areas seems adequate in particular when using a water allocation model. The synthesizing algorithm is efficient and can be easily applied in different regions. The model itself is mainly linear, even if it is corrected by yearly precipitation and an eventual saturation due to the extension of tree crops. In other situations, for example, a late expansion of irrigated areas, or where a reduction of the expansion is already noticeable, it will be necessary to switch to a non-linear system. A simple ordinary least square adjustment has been used to fit the coefficients, but other more sophisticated techniques could be used.

Those different questions are being taken into account in the frame of the AMETHYST project aver the Merguellil watershed in Tunisia

**5. The introduction seems somewhat disconnected in how the authors present their case, hypothesis, background, and approaches. Also, more detailed connections should be drawn between each of the paragraphs to help with the flow and motivation more than simply introducing some key concepts.**

Agree. Thank you for your suggestions, we have rewritten parts of the introduction. In response to your comment n°14, part of section 2.1 has been moved to the introduction. We feel that it helps to better understand the rationales of this study. We have also made a rewriting of some sentences. In particular, we modified the sentence that used the words "classic approach" and "curve fitting approach" that were too vague (see point 11).

The new introduction now makes a deeper overview of the south-Mediterranean situation and introduce the challenges of the Tensift watershed:

To satisfy the continuous increase in food demand associated with population growth, the agricultural sector has been asked to pursue its already initiated process of conversion towards agricultural intensification and above all towards a sharp increase in yields. This context goes hand in hand with the increaseee in food trade. The replacement of traditional crops by more financially attractive crops is already underway **(Jarlan et al., 2016)**. In the "growth" scenario (which is more or less the actual trend),  presented by **(Malek et al., 2018)**, the annual production of cultivated land increases by 40% and, the production of permanent crops increases by 260%. In the "sustainable" scenario, annual crop production and tree production increase by 30% and 38%, respectively. As a consequence, irrigation water needs are expected to increase **(Fader et al., 2016)**. The expansion and intensification of tree crops will also further rigidify the demand for agricultural water and increase the pressure on groundwater reservoirs **(Jarlan et al., 2016)** in order to keep tree crops alive during drought events **(Le Page and Zribi, 2019; Tramblay et al., 2020)**. This study is carried on in the Tensift basin in Morocco, where the increase in the irrigated area and the intensification of irrigation during recent decades have caused a long-lasting drop in the groundwater table **(Boukhari et al., 2015)**. A multi-model analysis of the area **(Fakir et al., 2015)** has shown that the groundwater table falls from 1 to 3 m/year and that the mean annual groundwater deficit (about 100 hm$^3$ since 2000) is equivalent to 50% of the reserves lost during the previous 40 years. Among the main causes of this depletion, is a reduction and higher irregularity of precipitation **(Marchane et al., 2017)** for crop growth and groundwater recharge, a reduction of snow water storage **(Marchane et al., 2015)**, an increase and intensification of irrigated areas, and a progressive conversion to arboriculture due to national strategy. Since irrigation relies increasingly upon groundwater abstraction, questions are inevitably raised concerning the future of local agriculture and groundwater.

We have also made a rewrite of the motivation of developing this approach regarding other techniques:

There is a significant amount of literature about the estimation of land use and land cover changes **(Mallampalli et al., 2016; Noszczyk, 2018)**, with various techniques to estimate or predict them. Many land cover change approaches are based on transition probability which was introduced by **(Bell, 1974)** and have been eventually connected to Cellular Automata to account for geographical interrelationships **(Houet et al., 2016; Marshall and Randhir, 2008)**. A very interesting technique has been to combine the top-down (demand-driven) and bottom-up (local conversion) processes of land cover change by proceeding to a simplification of local processes **(van Asselen and Verburg, 2013; Verburg and Overmars, 2009)**. Despite a huge bibliography both in climate change and land cover change, scenario analysis over the past 25 years has mostly focused on climate change projections, while the impact on land use and land cover has been neglected. **(Titeux et al., 2016)** found that

only 11% of the 2313 studies analyzed have included both land cover and climate changes. Also based on a large review, March et al. (2012) have called this a "hegemony" of climate as a driver of change. Furthermore, the implementation of land cover change techniques appear to be tedious and does not account for the intensification of cropping patterns. The motivation of the present work is to take into account both land cover change and crop intensification in future scenarios of irrigation water demand, taking into account the impact of climate change.

**6. Page 8, lines 225-230: This summary paragraph of the results related to Figure 4 refer to calibrated and validated years. However, in the previous paragraph in Section 4.1, the authors describe in the second sentence that the plots include calibration "over the entire time series".**

**Then there is Figure 5, which is never mentioned or hardly discussed in the results discussion, though some of the statistical results are discussed, e.g., such as the last sentence on page 8 (lines 230-231), for the 1/3 test. Please make sure to address these issues in a future submission, and that results are well described and figures are discussed appropriately.**

Agree. The purpose of this figure was not clear, and presenting the figure 4 which is computed over the whole time series before the evaluation (figure 5) was confusing. Also, the explanation of figure 5 had been accidentally deleted from the previous version of the manuscript, which mad it further confuse.

Now, the explanation of figure 5 comes first and is numbered Figure 4. It also explains the calibration/validation strategy:

Figure 4 shows the three statistical metrics of the fit for different calibration groups. The calibration is computed against a subset of the 16 years, where one half (1/2) means 8 calibration years for 8 validation years, one third (1/3) means 5 calibration years for 11 validation years and one quarter (1/4) means 4 calibration years for 12 validation years. $r^2$ were generally located around 0.5 which indicates an average fit, but it decreased when the calibration set size decreased. RMSE ranged between 0.1 for half of the calibration years to 0.17 for the corrected methods for only four calibration years. It must be noted that the Kcsim and Kccor versions degrade the model performances in terms of RMSE when only 1/3 or 1/4 of the years are used. This is also true for the dry calibration years which is explained by the fact that the rainfall correction (Kccor) needs good representativeness of extreme cumulative precipitations.

The explanation of the old figure 4 (now figure 5) now begins with:

The trajectory of each irrigated sector can be very different, according to its crop mixture, cropping density, and evolution. Figure 5 shows an example of this diversity with four contrasted irrigated areas. The figure also shows the calibrated and original time series of $K_c$ when the calibration is carried out over the entire time series.

**7. Abstract: The abstract text seems to skip critical information when relating the future climate scenarios and the use of the derived synthetic crop coefficients. For example, on line 19, the authors mention "loss of performance . . . is to be expected for two time periods after the observations (2050)". How does "2050" relate here to the "two time periods"? What are they?**

Agree. Thank you. The two time periods were referring to the fact that when sub-sampling the training dataset to one-third of the 16 available years, $r^2$ was reduced to 0.45. This score has been interpreted as the level of reliability that could be expected for two time periods after the full training years (thus near to 2050).

The abstract has been significantly rewritten in response to point 7, 8 and 9 below (see comment n°9 for the new version of the abstract)

**8. Then the authors mention "This flexible system of equations" – what system of equations? The authors have room to increase the content of the abstract to help the summary make sense for readers potentially interested in reading the article. However, it is hard to follow the summary of results and methods introduced by the current way it is written.**

Agree. The corresponding part of the abstract has been rewritten to make things clearer.

**9. Also, what are the "two agricultural scenarios" mentioned on line 22? Again, it would be useful to at least provide some introduction or link back to early in the abstract the ties when such references are made.**

In the context of major changes (climate, demography, economy, etc.), the Southern Mediterranean area faces serious challenges with intrinsically low, irregular, and continuously decreasing water resources. In some regions, the proper growth both in terms of cropping density and surface area of irrigated areas is so significant that it needs to be included in future scenarios. A method for estimating the future evolution of irrigation water requirements is proposed and tested in the Tensift watershed, Morocco. Monthly synthetic crop coefficients ($K_c$) of the different irrigated areas were obtained from a time series of remote sensing observations. An empirical model using the synthetic $K_c$ and rainfall was developed and fitted to the actual data for each of the different irrigated areas within the study area. The model consists of a system of equations that takes into account the monthly trend of $K_c$, the impact of yearly rainfall, and the saturation of $K_c$ due to the presence of tree crops. The impact of precipitation change is included in the Kc estimate and the water budget. The anthropogenic impact is included in the equations for $K_c$. The impact of temperature change is only included in the reference evapotranspiration, with no impact on the $K_c$ cycle. The model appears to be reliable with an average r2 of 0.69 for the observation period (2000-2016). However, different sub-sampling tests of the number of calibration years showed that the performance is degraded when the size of the training dataset is reduced. When sub-sampling the training dataset to one-third of the 16 available years, $r^2$ was reduced to 0.45. This score has been interpreted as the level of reliability that could be expected for two time periods after the full training years (thus near to 2050).
The model has been used to reinterpret a local water management plan and to incorporate two downscaled climate change scenarios (RCP4.5 and RCP8.5). The examination of irrigation water requirements until 2050 revealed that the difference between the two climate scenarios was very small (<2%), while the two agricultural scenarios were strongly contrasted both spatially and in terms of their impact on water resources. The approach is generic and can be refined by incorporating irrigation efficiencies.

**10. Page 2 – Lines 33-34: Please clarify a bit more here what is defined by "dynamics of irrigated areas is strong". What dynamics are specifically referring to, e.g., overexploitation of groundwater, changing climatic conditions, etc.?**

Yes. We agree that the word "dynamic" is indeed not explicit enough. In response to the reviewer's comment, it has been replaced by "cropping density and surface area of irrigated areas "

**11. Page 3 – Lines 65-69: Please more fully describe and support with previous studies the types of approaches (i.e., "classic approach" vs. "a curve fitting" one) that are mentioned here. The language used here is somewhat vague on the background of the different approaches.**

Ok, thank you. Both terms can be simply discarded to ease reading as follows:

Each synthetic time series would then account for the spatial variability of cropping patterns inside each irrigated area. Unless no sudden change occurs, a statistical model accounting for the monthly trend of Kc, the impact of rainfall, and the effect of saturating the land cover with tree crops should give an accurate fit that would allow extrapolating into the next few decades. As the synthesis of Kc for separated irrigated areas would also decrease

substantially the amount of information compared to a land cover approach, so that some information, like the amount of tree crops, should be retrieved back from the time series.

**12. Page 3 – Line 79: Add either a comma, or more appropriately a semicolon, and the word "and" between "climate change" and "3) an alternative scenario. . ."**

Thank you. Done.

**13. Page 3 – Line 86: The figure referenced here, "Figure 3", should be "Figure 1".**

Thank you. Corrected.

**14. Page 3-4, Lines 94-102: This last part of the paragraph (Section 2.1) seems more appropriate in the Introduction section as part of the motivation in conducting this study. Authors may want to consider moving part or most of this description to the Introduction.**

Thank you very much for this suggestion. You are right, we have now introduced the study case much earlier into the introduction and moved this paragraph as a rationale for the present study.

**15. Page 4, Lines 114-115: Which station surface network data were used to evaluation the GSWP3 forcing fields? Or are you citing values from another published reference here? Please further specify the source for this GSWP3 validation and results.**

OK. Agree. The evaluation os GSWP3 has been developped:

Statistical scores (bias and daily correlation coefficients) were computed between the GSWP3 product and 11 SYNOP stations over north-west Africa over the 1981-2010 period. GSWP3 well represents the variability of temperature relative humidity and total precipitation. The correlation coefficients computed between both monthly time series are 0.99, 0.87, and 0.74 for temperature, relative humidity, and precipitation, respectively. Standard deviations are quite small for the three variables (0.01, 0.03, and 0.12 respectively). At the daily time step, GSWP3 well represents the day-to-day temperature variability and especially the relative humidity. However, GSWP3 has more errors in terms of the daily evolution of precipitation (r=0.08, stdev=14.6). In semi-arid regions, where precipitation is infrequent and often convective, it is difficult to reproduce the precipitation events at the actual time. In terms of bias, the absolute temperature and relative humidity values are close to that observed: 0.22 °C and 0.54% respectively, meanwhile precipitation is underestimated by approximately 10% on average, but this bias is not always consistent from one station to another.

**16. Page 4, Line 117 (last sentence here): Which "meteorological dataset extends from 2000 to 2050", e.g., the downscaled RCP fields? Please further clarify this statement or move to a different location within this paragraph to accompany the appropriate data set description.**

Agree. In response to the reviewer comment, the sentence was rewritten as follows

The resulting downscaled and rescaled (to 1 km resolution) dataset of meteorological variables (Rain, Temperature, Wind Speed, Relative Humidity and Global Radiation) extends from 2000 to 2050

**17. Page 4, Line 118: Please spell out the full acronyms for "MODIS" and "NDVI".**

Thank you. Done.

**18. Page 5, Line 132: "small rainfalls" should simply be "small rainfall events . . .".**

OK, Done.

**19. Page 7, Line 179: "GCMs" is referred to here, but nowhere in the paper is this mentioned. Did the authors mean to refer to the "RCMs" or "RCPs"?**

Thank you, corrected with RCPs.

**20. Page 7, Lines 181-188: The authors indicate the years selected for calibration and validation, however, only the calibration years are noted here and how they are grouped. How are the validation years then selected separate from the calibration years? Please describe in more detail and clarify why the use of the different calibration year set groupings were selected.**

See response to point 6.

**21. Pages 7-8, lines 195-205: Please provide a bit more detail in how the upper limit of the crop-cover expansion is used to "bend the curve" and derivation of the coefficient, bc, is estimated.**

Agree. In response to the reviewer's comment, the "alternative scenario" was detailed and the choice of the coefficients are now argued as follows:

The main drivers defining these changes include 1) an expansion of irrigated areas through groundwater pumping; 2) the conversion of surface irrigation to drip irrigation; 3) intensification of arboriculture, and 4) an abandonment of irrigated cereals for vegetable crops. However, neither the plans nor the interviews specify the intensity or locations of these changes. Here, we do not address all of the measures considered, only those concerning land-cover changes. A reduction in the rate of increase of olive trees areas by 50% is expected to begin in 2021. At the same time, an intensification due to intercropping and the replacement of traditional Moroccan Picholine (100 to 200 trees per hectare) by Spanish Arbequina with a density of 1000 trees per hectares will take place. This is correlated with the expected production increase of 27%. The fate of citrus trees is less clear. The recent trend is a strong expansion, however, the plan expects stabilization of citrus orchards due to the lack of water especially in the western part of the basin. Finally, a progressive reduction of cereal crops (-722 ha/year) is expected.

**22. Page 8, line 236: Why would Kc values "plummet in the rare wet years" and only for the Mejjat region? Wouldn't more rain simply mean less irrigated water demand but Kc values remain up? Please explain further why this occurs.**

Our apologies for the error in this sentence. You are right, more rain means less irrigated water demand, and also a stronger development of the crops. The sentence is corrected to:

A boom of groundwater exploitation has provoked the development of irrigated areas. $K_c$ spikes in the rare wet years, but most of the time, the amplitude of the signal is very small, showing that the area is mostly composed of bare soils and tree crops.

**23. Page 8, line 238: "XX century" – do the authors mean at the beginning of the 21st Century, starting around 2000? Please check what was is intended here.**

Right. This has been corrected … we are into the 21st Century

**24. Page 9 + Figure 6: The results described here in relation to Figure 6 make many assumptions about how future crop coefficient values and irrigated scenarios play out under the different climatic scenarios and alternative Kc approach. The authors describe some of the results, but there appears to be some issue with how a value approaches certain constant values, such as for Ourika,**

**that it makes it somewhat difficult to understand the results fully. The authors may want to revisit whether these results are actually robust and update their description here, as needed.**

Agree. Thank you. The main assumption is that future crop coefficients will behave in the same way as today (crop cycle, relation between Kc and ET) while the fertilizing effect of CO2 could shift the growth cycle and shorten its length (see Bouras et al., 2019 among others). For the latter, an alternative approach based on interactive plant growth was proposed (Garrigues et al., 2015). The method also assumes that the irrigation framework does not change, when it will very likely modernize for example by shifting from gravity irrigation toward localized irrigation. This process is already started in response to an ambitious public policy called the "Plan Vert" for the Moroccan agriculture.

Equation 14 and figure 8, can be tuned to take into account the evolution of the irrigation framework. In response to the reviewer's comment, it is now clearly stated in the new version of the manuscript as follows:

Finally, the assessment of irrigation water demand should also take into account different losses at the system level (storage, transportation and operating losses) and at the plot level (deep percolation and runoff). The future evolution of the irrigation framework toward more efficient systems such as drip could also be considered using equation 14.

Concerning the issue of how $K_c$ approaches a constant value, the result was expected but it was probably not properly explained in the old version of the manuscript. Indeed, the increase of the crop coefficient (equation 8) is limited by the proportion of tree crops in an area. Therefore, when an area is fully covered with tree crops, meaning that there is no annual crop anymore, the synthetic $K_c$ has no seasonal evolution and is equal to the one of the tree crop (0.55 for Olive orchard in our case as stated on page 6, line 168). In ease the understanding, the sentence on lines 241-242 has been rewritten as follows:

The full coverage with tree crops was attained around 2040 in the Ourika area. Therefore, the synthetic Kc of this area is controlled by the maximum Kc value of equation 8, which was set as to the Kc of olive trees (0.55) which is the dominant tree crop of this region.

**25. Page 9, line 240: Change "feed" to "fed".**

Ok. Done.

**26. Figure 2: Make sure "NDVI" is consistently used throughout the figure. For example, noticed "Ndvi" in the "Empirical Relationships" container should be capitalized. Also, in the caption, "parenthesis" should be changed to "parentheses".**

Thank you. Corrected.

**27. Figure 3, title and caption: It would be helpful if details of which irrigated area, crop type, and month were highlighted in this example figure and caption.**

OK. The beginning of the caption has been modified to:

Quantities used to perform the simulation. The example illustration on the left depicts the synthetic Kc of the N'Fis private area where a mix of tree crops (olive, orange, apricot) and cereals crop (mainly wheat) are cultivated. The peak occurs during March/April when cereal crops are at their maximum development, the valleys correspond to the summer months when there are no seasonal crops.

**28. Figure 4: Do these plots reflect the simulated Kc values of Equation 9? Authors may want to specify that here in the caption.**

OK. The caption has been modified to:

Examples of $K_c$ simulations (eq.9) compared to the original time series for four selected areas.

**29. Figure 6: It is difficult to distinguish the four grey lines from each other in this plot and what is conveyed in the legend. It might be better to switch or replace some of the time series with colored lines.**

That's true, the figure has been redrawn with colored lines.

**30. Figure 8: "gravitary"? Did the authors intend to indicate "gravity" irrigation type? Authors might want to indicate that the graphic is showing "cumulative" IWR and rainfall for the two year period.**

Thank you for both recommendations. "gravitary" has been replaced by "surface".

The title of the figure and the caption of his figure have been corrected as suggested.